# Three Months’ PSA and Toxicity from a Prospective Trial Investigating STereotactic sAlvage Radiotherapy for Macroscopic Prostate Bed Recurrence after Prostatectomy—STARR (NCT05455736)

**DOI:** 10.3390/cancers15030992

**Published:** 2023-02-03

**Authors:** Giulio Francolini, Pietro Garlatti, Vanessa Di Cataldo, Beatrice Detti, Mauro Loi, Daniela Greto, Gabriele Simontacchi, Ilaria Morelli, Luca Burchini, Andrea Gaetano Allegra, Giulio Frosini, Michele Ganovelli, Viola Salvestrini, Emanuela Olmetto, Luca Visani, Carlotta Becherini, Marianna Valzano, Maria Grazia Carnevale, Manuele Roghi, Sergio Serni, Chiara Mattioli, Isacco Desideri, Lorenzo Livi

**Affiliations:** 1Radiation Oncology Unit, Oncology Department, Azienda Ospedaliero Universitaria Careggi, 50134 Firenze, Italy; 2Department of Biomedical, Experimental and Clinical Sciences “M. Serio”, University of Florence, 50121 Firenze, Italy; 3CyberKnife Center, Istituto Fiorentino di Cura e Assistenza (IFCA), 50134 Firenze, Italy; 4Department of Urologic Robotic Surgery and Renal Transplantation, Careggi Hospital, University of Florence, 50121 Firenze, Italy

**Keywords:** radical prostatectomy, adjuvant therapy, local invasion

## Abstract

**Simple Summary:**

This article is a report about early toxicity and biochemical outcomes after stereotactic salvage radiotherapy for macroscopic recurrence within prostate bed after radical prostatectomy. Data reported suggest optimal tolerability profile and promising oncologic outcomes after this approach within a prospective multicentric trial.

**Abstract:**

Biochemical recurrences after radical prostatectomy (RP) can be managed with curative purpose through salvage radiation therapy (SRT). RT dose escalation, such as stereotactic RT (SSRT), may improve relapse-free survival in this setting. STARR trial (NCT05455736) is a prospective multicenter study including patients affected by macroscopic recurrence within the prostate bed after RP treated with SSRT. Recurrence was detected with a Choline or PSMA CT-PET. In the current analysis, the early biochemical response (BR) rate and toxicity profile after three months of follow-up were assessed. Twenty-five patients were enrolled, and data about BR and toxicity at three months after treatment were available for 19 cases. Overall, BR was detected after three months in 58% of cases. Four G1–G2 adverse events were recorded; no G ≥ 3 adverse events were detected. SSRT appears feasible and safe, with more than half of patients experiencing BR and an encouraging toxicity profile. The STARR trial is one of the few prospective studies aimed at implementing this promising treatment strategy in this scenario.

## 1. Introduction

Radical Prostatectomy (RP) is one of the preferred treatment approaches for localized prostate cancer (pCA). However, up to 29% of men undergoing RP eventually develop recurrence within 10 years of surgery [1]. Prostate bed salvage radiotherapy (SRT) is a widely accepted treatment option for this scenario [2]; nonetheless, the outcome after salvage treatment is significantly worse in patients with positive metabolic imaging detecting nodal or distant macroscopic recurrences [3]. In this scenario, RT dose escalation aimed at improving disease control is advocated by some authors [4,5], but no consensus exists on the management of macroscopic relapse detected within the prostate bed. According to European Association of Urology (EAU) guidelines, given the variations of techniques and dose-constraints, a satisfactory agreement about target volume definition and optimal SRT dose has not been well defined [6]. Modern imaging methods (e.g., Choline or PSMA PET/CT and Magnetic resonance imaging-MRI) allowed the precise definition of the extent and location of recurrence within the prostate bed, and prompted the use of tailored treatment approaches including Stereotactic SRT (SSRT) on macroscopic relapse. Currently, SSRT is considered experimental and should be restricted to clinical trials [7], but promising results have been reported in preliminary experiences, even in comparison to conventional SRT [8]. SSRT yields potential advantages if compared to conventional SRT considering the dose-escalated approach on a limited treatment volume, deliverable in a lower number of fractions (usually 3–5). Given the lack of prospective evidence about this issue, a prospective trial was designed, enrolling patients affected by macroscopic prostate bed relapse undergoing SSRT (STereotactic sAlvage Radiotherapy for macroscopic prostate bed Recurrence after prostatectomy, STARR trial, NCT05455736). STARR trial is aimed at prospectively assessing the rate of biochemical relapse and adverse events after SSRT in this setting. In the present work, we present an analysis of the first cohort enrolled within the trial, focusing on early biochemical outcomes and acute toxicity.

## 2. Materials and Methods

### 2.1. Study Population

STARR (NCT05455736) is a prospective multicenter study including patients treated with RP for localized prostate cancer and affected by macroscopic recurrence within the prostate bed. Recurrence was detected with a Choline or PSMA CT PET performed at biochemical recurrence (PSA > 0.2 ng/mL) after surgery, and confirmed with MRI (except for patients in whom MRI was contraindicated). No imaging after surgery and before biochemical relapse was routinely performed to rule out the presence of residual prostate gland tissue. All patients with evidence of regional or distant metastatic disease were excluded from the trial. All patients in whom any contraindication to SSRT was detected (e.g., chronic bowel inflammatory disease, relevant toxicity after surgery) were excluded from the trial. Patients reporting urethral stenosis after surgery or relevant incontinence were not deemed fit for SSRT and were excluded from the trial.

### 2.2. Study Procedures

All patients underwent Cyberknife^R^ SSRT for a total dose of 35 Gy in 5 fractions. Treatment was administered with an every-other-day schedule. Use of different techniques (e.g., Volumetric Modulated Arc Therapy-VMAT or MR based SSRT) was allowed within the protocol, provided that the treatment respected doses and fractionation indicated per protocol (35 Gy in 5 fractions) and that dose constraints to organs at risk were observed. However, first, patients were enrolled in the promoting institution, where CyberknifeR SSRT is used for this kind of treatment. PSMA or choline CT-PET and MRI were co-registered with the planning CT scan for target volume delineation purposes. The Gross Target Volume (GTV) corresponded to macroscopic neoplastic tissue within the prostate bed. Clinical Target Volume (CTV) was obtained, adding a 2 mm margin to GTV. A margin of 3 mm (1 mm in the posterior direction) was added to CTV to obtain the final Planning Target Volume (PTV) (Figure 1). Patient alignment and target tracking were performed through implanted fiducials. During delivery, radiographic images were acquired using the InTempo^TM^ System (AccurayInc.), which alters imaging frequency between 15 and 60 s depending on the magnitude of the prostate or recurrence on prostatic bed motion detected. Bladder catheter placement was not performed for planning and delivery procedures. The following organs at risk were contoured: rectum, bladder, bowel, urethra, penile bulb, femoral heads, and bowel. The main dose constraints used are reported in Table 1. Concomitant Androgen Deprivation Therapy (ADT) was not allowed (patients should be free from ADT from at least 12 months before enrollment). All patients were evaluated every three months with a PSA and a clinical examination. The first treatment response evaluation was performed at 3 months after end of treatment with serum PSA. No protocol assessments were performed before 3 months from end of treatment. Re-staging was performed in case of biochemical or clinical progression of disease.

### 2.3. Outcomes

Complete biochemical response and biochemical response were defined as a PSA nadir ≤ 0.2 ng/mL and ≤50% of baseline, respectively. Biochemical relapse was defined as a PSA increase above 0.2 ng/mL for patients with a PSA nadir ≤ 0.2 ng/mL (or 2 consecutive PSA increases > 25% if compared to nadir in patients with a PSA nadir > 0.2 ng/mL). These definitions were adapted from the Prostate Cancer Working Group 3 recommendations [9]. A PSA was defined as stable if neither a biochemical response nor biochemical relapse could be defined. Acute Gastrointestinal (GI) and Genitourinary (GU) Toxicity was assessed every 3 months after treatment according to the Common Terminology Criteria for Adverse Events (CTCAE) score v.4.03 [10].

### 2.4. Statistical Analysis

Primary endpoint of the trial will be to assess rate of biochemical relapse free patients after 2 years of follow up. Considering previous reported data about use of SSRT in this setting [11], a sample size of 90 patients will be needed to assess with a +/−9% margin of error the biochemical relapse free survival rate in this population. In this work, we present early results in terms of acute toxicity and early biochemical outcomes after 3 months from treatment in the first cohort of enrolled patients.

## 3. Results

As of 25 September 2022, twenty-five pts have been enrolled from March 2021. Data about biochemical response and toxicity at 3 months after treatment were available for 19 of these patients, included in the current analysis. The main features of the included population are summarized in Table 2. The median time from surgery to recurrence was 37 months (IQR 21.7–124.5). The median PSA at recurrence was 1.13 ng/mL (IQR 0.43–2.3 ng/mL). Macroscopic recurrence was detected by PSMA PET/CT or Choline PET/CT in fifteen (79%) and four (21%) patients, respectively. Five patients did not perform MRI confirmation. PET/CT and MRI imaging were reviewed by nuclear medicine and a dedicated radiologist with more than 5 years of experience in genitourinary cancers.

Overall, complete biochemical response and biochemical response were detected after 3 months in five (26.3%) and eleven (58%) of cases, respectively. One biochemical relapse at first evaluation has been registered and the patient started ADT; seven patients had stable PSA if compared to baseline and continued observation. A PSA reduction was detected in 17 patients, with a median PSA drop (defined as difference between PSA at baseline and PSA after 3 months from treatment) of 0.53 ng/mL (IQR 0.17–1.59) (Figure 2). None of the patients reported GI or GU symptoms at baseline (after surgery and before SSRT). Adverse events were recorded in two patients, one patient reported G2 GI and G1 GU toxicity, and G2 GU and GI toxicity were reported in another case.

## 4. Discussion

### 4.1. Clinical Outcomes

In the current analysis, promising biochemical results after SSRT in patients affected by macroscopic recurrence were evidenced, with a favorable toxicity profile. Biochemical response was reported in more than half of patients, and 26% of them reached a complete biochemical response within 3 months of treatment. These promising results in terms of biochemical outcomes, despite the unfavorable prognostic impact of macroscopic disease within the prostate bed [3], may have been related to the possibility of a dose escalated treatment. Indeed, considering an alpha/beta ratio for prostate cancer of 1.5 or 3 Gy [12], the dose fractionation schedule in the used STARR trial would correspond to a 2 Gy equivalent dose (EQD2) of 85 or 70 Gy, respectively, which is considerably higher if compared to a standard conventional postoperative treatment delivering 64–66 Gy [2].

### 4.2. ADT Administration

ADT was not allowed by protocol, meaning that biochemical outcome was related to local treatment alone. Of course, some clinicians may feel that short course concomitant ADT should be performed in a similar scenario. However, trials suggesting benefits of concomitant administration of ADT in this setting (e.g., GETUG AFU 16 or RTOG 9601 [13,14]) were conducted in patients in whom metabolic imaging was not performed before salvage treatment, suggesting that part of the benefit could be related to the spatial cooperation between local and systemic treatment. In our opinion, the use of local treatment alone would be justified in a prospective trial conducted on patients with a more precise staging aimed to exclude subclinical disease outside the prostate bed. However, the majority of patients (74%) were affected by disease with ISUP scores ≤ 3, and the likelihood of subclinical microscopic disease is lower in a similar population if compared to Gleason 8 or higher prostate cancer. For this reason, escalation in terms of local treatment might have been effective even in the absence of concomitant ADT administration.

### 4.3. Current Status of PET-Directed sRT

Tailoring the treatment volume on the basis of PET/CT and MRI allowed a reduction of the overall treatment volume and the profitably spare organs at risk. Interestingly, no data about regional, distant or local relapse detection rate with PSMA or Choline imaging are available from this series, because patients with regional or metastatic disease were excluded from the trial and screening failure procedures were not recorded. Insights about these issues may be related to early PSMA detection results reported within a parallel project running in our center (PSICHE trial, NCT05022914). Of note, further reduction in terms of acute toxicity may be related to the use of the Cyberknife^R^ robotic stereotactic technique, as suggested by a post hoc analysis of a PACE-B trial [15]. Thus, SSRT may have the potential to increase the benefit-to-risk ratio in the complex scenario of postoperative recurrence. Nevertheless, salvage radiotherapy is currently a debated issue, especially regarding correct prognostic stratification and the addition of concomitant ADT [14,16]. However, many of the pivotal trials published in this scenario were conducted on conventional imaging only, while recent evidence confirmed the significant impact that sensitive imaging methods may have on postoperative management. The availability of modern imaging and RT techniques allows the precise identification of the site and extension of disease, prompting the development of treatment strategies aimed at maximizing clinical outcomes. The EMPIRE-1 trial was a single center phase 2/3 trial including patients with a detectable PSA after RP, randomized to receive conventional imaging alone or with 18F-fluciclovine-PET/CT. In the experimental group, radiotherapy management and target delineation were determined by PET findings. The trial enrolled 165 patients, and results showed a significant advantage for patients undergoing next generation imaging in terms of three-year event free survival (75.5% vs. 63%, *p* = 0.002), and a similar toxicity in both study groups. Authors concluded that next generation imaging significantly improved survival free from biochemical recurrence or persistence, and that a novel PET radiotracer should be integrated into radiotherapy decisions [17]. Of note, the EMPIRE-1 trial did not included stereotactic radiotherapy within a pre-determined management algorithm, while other ongoing trials (e.g., PSICHE trial, NCT05022914) are currently implementing such treatment strategy within a PSMA guided framework. Despite the emerging role of next generation imaging and stereotactic RT, no standardized approach for patients with pelvic macroscopic evidence of disease detected after a postoperative PSA rise currently exists, and various strategies have been proposed.

### 4.4. Dose-Escalated Radiotherapy in Salvage Setting and Comparison with Other SSRT Series

Dose-escalated conventional radiotherapy was proposed within various retrospective studies, reporting biochemical progression free survival ranging between 44 and 89% and late G3 GU or GI toxicity ranging between 2 and 7.3% [4,6,18,19,20,21,22]. SSRT has been proposed as an alternative in this scenario in one previous retrospective series of 90 patients, reporting an overall biochemical free survival rate of 72% after an average follow up of 21.2 months and no G > 2 adverse events reported [11]. However, that series included both patients treated with Cyberknife^R^ robotic technique and with intensity modulated RT (IMRT) using the VERO^R^ system, while the present analysis is based only on Cyberknife^R^ treated patients. This makes a direct comparison difficult; still, it suggests that SSRT may be feasible as well with various RT techniques, expanding this treatment possibility to different facilities experienced in stereotactic radiotherapy. Of note, this approach was compared within a propensity score matched analysis with conventional salvage radiotherapy. In brief, data from 185 patients treated in seven Italian centers for macroscopic prostate bed recurrence were retrospectively collected. After propensity matching, 90 patients in the conventional and SSRT group were selected and compared (45 in each group). Results did not show any significant difference in terms of biochemical relapse free and progression free survival. However, a lower rate of toxicity was evidenced for patients treated with SSRT, with acute GI and GU adverse events reported in 4.4 versus 44.4% (*p* < 0.001) and 28.9 versus 46.7% (*p* = 0.08) of patients, and late GI and GU adverse events reported in 0 versus 13.3% (*p* = 0.04) and 6.7 versus 22.2% (*p* = 0.03) of patients, respectively [8]. SSRT to prostate bed was tested, as well, in two prospective phase I trials [23,24]. These demonstrated the feasibility of dose escalation up to 35–45 Gy without dose limiting toxicity, confirming the good safety profile of ultrahypofractionated treatment for postoperative RT. Ballas et al. published a phase I dose escalation trial aiming to evaluate the maximum tolerated dose after increasing hypofractionation to the prostate bed. Authors tested three dose levels (3.6 Gy × 15 fractions, 4.7 Gy × 10 fractions and 7.1 Gy × 5 fractions) on a twenty-four patient cohort with at least 6 months of follow-up. Results showed that no G ≥ 3 GI or GU toxicity was seen at any dose level. Seven of twelve patients enrolled in the 7.1 Gy × 5 fractions cohort experienced a G2 GI toxicity during treatment, and one out of twelve patients of the same group had an increase to G1 and G2 GU toxicity in the two weeks after RT. Moreover, 71% of patients had a minimally important difference in terms of Expanded Prostate Cancer Index Composite score bowel domain at week 2 after treatment, while International Prostate Symptom Scores worsened two weeks after treatment but improved by six to ten weeks. Authors concluded that long-term follow up was needed after SBRT due to the transient G2 increase in rectal toxicity occurring during and immediately after radiotherapy [23]. Sampath et al. published another dose escalation trial including patients with organ-confined, node-negative prostate cancer who had biochemical failure after prostatectomy with a PSA ≤ 2 ng/mL. In their cohort, the dose escalation protocol provided treatment with 35 Gy, 40 Gy and 45 Gy in five fractions administered on alternate days. After the enrollment of 26 patients, the median follow up was 60, 48 and 33 months in the 35, 40 and 45 Gy cohort, respectively. Results reported that no acute dose limiting toxicity events were observed, while late G ≥ 2 and ≥ 3 GI toxicity was reported in 11% and 0%, respectively, and 38% and 15% of patients suffered late grade ≥ 2 and ≥ 3 GU toxicity, respectively. Interestingly, no increase in terms of late GU toxicity was reported when comparing the 45 Gy to the 40 Gy cohort, and the crude rate of complete biochemical response was 42% [24]. Authors concluded that the recommended dose for a phase 2 study should have been 40 Gy in five fractions, which is slightly higher than the dose proposed within the STARR trial. However, a comparison between these two studies and the STARR trial is difficult because both of them included all the prostate bed within treatment volume, and included patients affected by biochemical recurrence only. From a safety reporting view, only patients with prostate bed relapse were enrolled in the STARR trial, with the target including only macroscopic evidence of disease and a lower extent of treatment volumes. Thus, the rate of adverse events in those Phase I trials may be negatively affected by larger target volumes, and the prognosis of patients enrolled within the STARR trial could be considered different due to the presence of macroscopic recurrence within the prostate bed. Despite the extreme heterogeneity of treatment approaches proposed, some sort of treatment intensification for these patients appears to be beneficial, as evidenced by a multicentric retrospective experience published in 2022, the SPIDER 01 study [25]. In this series, authors collected data about 363 patients treated in 16 European centers for biochemical recurrence and prostate bed macroscopic relapse within the prostate bed proven by functional imaging. Patients were treated between January 2000 and December 2019 and divided into four groups according to the delivered treatment (dose escalation on macroscopic recurrence, dose escalation on prostate bed, dose escalation on prostate bed and macroscopic recurrence, no dose escalation). After a median follow-up of 53.6 months, five-year progression free survival and metastasis free survival were 70% and 83.7%, respectively, with rate of G ≥ 2 GU and GI late toxicity of 12 and 3%, respectively. Of note, results showed a five-year progression free survival benefit for all groups with any dose escalation > 72 Gy (72.8% vs. 60.3%, *p* = 0.03). Authors concluded that the integration of functional imaging in the salvage treatment approach is effective when macroscopic relapse is detected inside the prostate bed, and that dose escalation had a significant impact on progression free survival. Thus, treatment intensification for patients with macroscopic prostate bed recurrence appears justified. Another ongoing prospective trial (SHORTER, NCT04422132) is currently enrolling patients randomized to receive either moderate hypofractionation (55 Gy in 20 fractions) or SSRT (32.5 Gy in 5 fractions) to the prostate bed +/− pelvic nodes, aiming to compare the rate of GI and GU symptoms after the two treatment approaches. Of course, the inclusion of prophylactic pelvic volumes in a five-fractions salvage treatment strategy would be an interesting field of debate, given the necessity to correctly assess the risk-to-benefit ratio in this particular scenario.

### 4.5. Potential Advantages of SSRT

SSRT on macroscopic appears an attractive approach due to the limited number of fractions and the potential favorable risk-to-benefit ratio. Altered fractionation (e.g., moderate hypofractionation) is already shown to be effective and feasible within this setting [26], but prospective evidence about more extreme fractionations (e.g., >5 Gy per fraction) is eagerly awaited. Of course, the postoperative management of prostate cancer in the era of next generation imaging is a complex issue, especially when pelvic disease has been detected. Different approaches exploiting the current advances in radiation therapy have been proposed [27], and real-world data confirmed that no increased or unexpected toxicity were detected after the use of hypofractionated regimens in this scenario [28]. Stereotactic radiotherapy constitutes a tailored approach aimed to treat macroscopic evidence of disease, avoiding wide prophylactic treatment volumes, and many prospective trials are currently testing this treatment strategy in different settings of prostate cancer treatment [29]. Furthermore, salvage radiotherapy guided by imaging has been shown to be effective in clinical practice [30]. The STARR trial (NCT05455736) is focused on implementing stereotactic radiotherapy in a well-selected cohort of patients affected by macroscopic prostate bed relapse. If the promising data reported in this early work are confirmed in a complete cohort, SSRT may represent an interesting treatment option allowing the performance of an effective salvage treatment in a low number of fractions. This could be particularly beneficial for patients’ quality of life (one of the secondary endpoints of the STARR trial) and facilities’ waiting lists. Moreover, longer treatments may be unhelpful in special situations, when hospital admittance is problematic (e.g., pandemics) [31]. Of note, concomitant ADT was not provided within the STARR trial, and biochemical outcomes in this early cohort are exclusively related to SSRT effect. This allows a reliance on clinical benefit from the curative local treatment, but spatial cooperation with ADT may improve the benefit in selected patients with adverse prognostic factors. The attitude of clinicians towards ADT is often heterogeneous [32], especially in a postoperative setting, and its benefit when SSRT is provided will be an interesting matter of debate.

## 5. Conclusions

Early results from the first cohort of patients enrolled within the STARR trial (NCT05455736) show promising biochemical outcomes and a favorable toxicity profile. Once completed, the STARR trial could constitute one of the first prospective pieces of evidence about treatment tailoring and intensification for patients affected by postsurgical relapse and macroscopic evidence of disease within the prostate bed. After the advent of modern RT techniques, and supported by the literature data, hypofractionation in the postoperative scenario is currently the mainstay approach for different pathologies (e.g., breast cancer) [33,34], and we advocate for the implementation of similar treatment strategies in prostate cancer. SSRT may represent a paradigm for the integration of novel imaging methods and modern RT techniques in routine clinical practice.

## Figures and Tables

**Figure 1 cancers-15-00992-f001:**
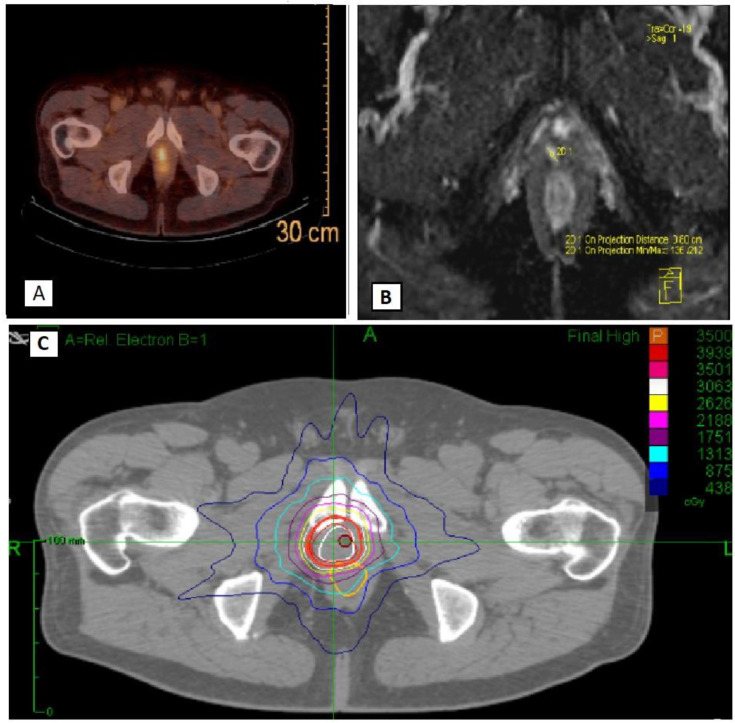
An example of radiotherapy treatment plan with PSMA imaging (**A**), MRI imaging (**B**) and Cyberknife^R^ treatment volumes and isodose lines (**C**).

**Figure 2 cancers-15-00992-f002:**
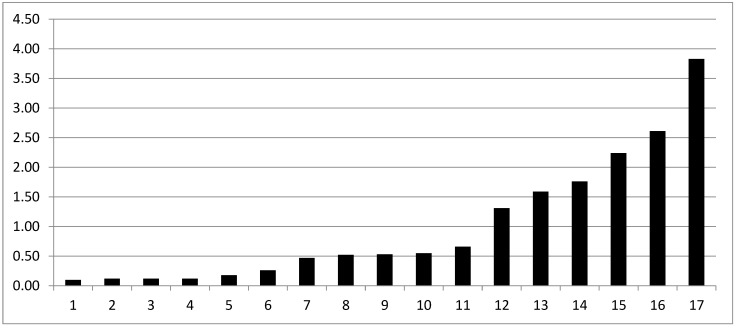
PSA drop for the 17 patients reporting any reduction in PSA. On the vertical axis, absolute values of PSA in ng/mL are reported.

**Table 1 cancers-15-00992-t001:** Main dose constraints used.

Organ at Risk	Dose Constraint	Aim
Rectum	V18.1 Gy V29 Gy V36 Gy	<50% <20% <1 cc
Bladder	V18.1 Gy V37 Gy	<40% <10 cc
Urethra	V42 Gy	<50% (not mandatory)
Femoral heads	V14.5 Gy	<5%
Penile bulb	V29.5 Gy	<50%
Bowel	V18.1 Gy V30 Gy	<5 cc <1 cc

**Table 2 cancers-15-00992-t002:** Principal baseline features of included patients.

Age (Median Value, IQR)	74 (IQR 69–80)
Baseline T stage (%)	T2b-c: 8 (42%) T3a-b: 11 (58%)
Baseline N stage	N0: 12 (63%) N1: 0 (0) Nx: 7 (37%)
Margin status	R0: 7(37%) R1: 12 (63%)
Baseline ISUP pattern	≤3: 14 (74%) Gleason 3 + 3:1 Gleason 3 + 4: 8 Gleason 4 + 3: 5 >3: 5 (26%) Gleason 4 + 4:4 Gleason 4 + 5:1
Baseline PSA (median, IQR)	1.13 ng/mL (IQR 0.4–2.3)
Baseline NCCN risk category	Low: 0 (0) Intermediate: 6 (32%) High: 13 (68%)

Note: Nx is defined as no lymph nodes removed.

## Data Availability

The data presented in this study are available on request from the corresponding author. The data are not publicly available due to privacy policy.

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
