# Peer review of "Three Months’ PSA and Toxicity from a Prospective Trial Investigating STereotactic sAlvage Radiotherapy for Macroscopic Prostate Bed Recurrence after Prostatectomy—STARR (NCT05455736)"

_cancers, 2023, doi:10.3390/cancers15030992_

Round 1
Reviewer 1 Report
-Cyberknife alone was used in this trial. Any reason not to use Linac based SSRT, or MR Linac / proton based SSRT?
-Can you please further define the contraindication to SSRT in this study, particularly regarding “relevant toxicity after surgery”?
-What were the baseline GI / GU symptoms following surgery (before SSRT)?
-Was the first assessment of GI/GU toxicity at 3 months after completion of the SSRT? How about assessment sooner than 3 months after completion such as at 1 month follow up?
-Can you please further explain the assessment of the treatment response such as when do you obtain the first PSA following SSRT?
Author Response
-Cyberknife alone was used in this trial. Any reason not to use Linac based SSRT, or MR Linac / proton based SSRT?
Answer: Use of different techniques (e.g Volumetric Modulated Arc Therapy-VMAT or MR based SSRT) is allowed within the protocol, provided that provided that the treatment respected doses and fractionation indicated per-protocol (35 Gy in 5 fractions) and that dose constraints to organs at risk were observed. However, first patients were enrolled in the promoting institution, where CyberknifeR SSRT is used for this kind of treatment. A comment was added in the section 2.2
-Can you please further define the contraindication to SSRT in this study, particularly regarding “relevant toxicity after surgery”?
Answer: Patients reporting urethral stenosis after surgery or relevant incontinence were not deemed fit for SSRT and were excluded from the trial. A comment was added in section 2.1
-What were the baseline GI / GU symptoms following surgery (before SSRT)?
Answer: None of the patients reported GI or GU symptoms at baseline (after surgery and before SSRT). A comment was added in section 3
-Was the first assessment of GI/GU toxicity at 3 months after completion of the SSRT? How about assessment sooner than 3 months after completion such as at 1 month follow up?
Answer: No protocol assessments were performed before 3 months from end of treatment. A comment was added in section 2.2
-Can you please further explain the assessment of the treatment response such as when do you obtain the first PSA following SSRT?
Answer: First treatment response evaluation was performed at 3 months after end of treatment with serum PSA
Reviewer 2 Report
It's an interesting and relatively well-written manuscript. I have several questions/comments:
- L34-L36: I find the statement is not completely true because in this well-known publication (ref 3), the authors distinguished high treatment response to SRT (negative or fossa-confined PSMA) versus men with poor response to SRT (nodes or distant-disease PSMA). The poor outcome after salvage treatment was worse for those with nodes or distant-disease PSMA and not local recurrence. Please reformulate.
-L37: "No consensus". Could you cite the lasted EAU recommendations to justify this?
- 57 : Please describe precisely the way you defined "macroscopic recurrence within prostate bed".
In fact, how many of them had PSMA PET and CHOLINE PET respectively? The PET results were retrived only from reports or there was (blinded) reviews of exams? By experieced nuclear medicine physicians (how many years of experience)? For the confirmation by MRI, how many didn't have the it because of the contradication? Please develop and explicit it. Also please describe succinctly the MRI procedure (radiologists' experience ect...)
- L60: exclusion for those had regional ou distant metastatic diesease: how many of them were detected by PSMA PET and CHOLINE PET respectively? We all know that their sensitivities are quite different. Please describe and discuss it in the discussion.
- L74. were conducted
- L84-L87: for the definition of complete biochemical response, biochemical response and biochemical relapse. It seems they were subjective choices. Are there data in the literature to support it? Please add it.
-L100:pts (patients). "At time of this analysis" is not very clear. Please define a date.
-In the table 1: There were mainly ISUP <or= 3, globally good pronostic. Please discuss it in the discussion.
-It would be good the have more figures, especially one showing the lesion of PET and MRI.
- there were some typing mistakes such as L156 (NCT NCT), L163 (4,17-21,5), L166 (G2 adverse). It is definitively not exhaustive. Please correct them. Otherwise, it give a neglected impression and that's a shame as it is an interesting manuscript...
Author Response
It's an interesting and relatively well-written manuscript. I have several questions/comments:
- L34-L36: I find the statement is not completely true because in this well-known publication (ref 3), the authors distinguished high treatment response to SRT (negative or fossa-confined PSMA) versus men with poor response to SRT (nodes or distant-disease PSMA). The poor outcome after salvage treatment was worse for those with nodes or distant-disease PSMA and not local recurrence. Please reformulate.
Answer: sentence was reformulated as following: “, nonetheless, outcome after salvage treatment is significantly worse in patients with positive metabolic imaging detecting nodal or distant macroscopic recurrences [3].
-L37: "No consensus". Could you cite the lasted EAU recommendations to justify this?
Answer: Citation to EAU Recommendations was added.
- 57 : Please describe precisely the way you defined "macroscopic recurrence within prostate bed".
In fact, how many of them had PSMA PET and CHOLINE PET respectively? The PET results were retrived only from reports or there was (blinded) reviews of exams? By experieced nuclear medicine physicians (how many years of experience)? For the confirmation by MRI, how many didn't have the it because of the contradication? Please develop and explicit it. Also please describe succinctly the MRI procedure (radiologists' experience ect...)
Answer: Macroscopic recurrence was detected by PSMA PET/CT, Choline PET/CT 15 and 4 patients, respectively. Five patients did not perform MRI confirmation. PET/CT and MRI imaging were reviewed by a nuclear medicine and a dedicated radiologist with more than 5 years of experience in genitourinary cancers. A comment was added expliciting these points.
- L60: exclusion for those had regional ou distant metastatic diesease: how many of them were detected by PSMA PET and CHOLINE PET respectively? We all know that their sensitivities are quite different. Please describe and discuss it in the discussion.
Answer: no data about regional, distant or local relapse detection rate with PSMA or Choline imaging are available from this series, because patients with regional or metastatic disease were excluded from the trial and screening failure procedures were not recorded. Insights about these issue may be related to early PSMA detection results reported within a parallel project running in our centre (PSICHE trial, NCT05022914). A comment was added in the discussion section
- L74. were conducted
Answer: sentence was rephrased
- L84-L87: for the definition of complete biochemical response, biochemical response and biochemical relapse. It seems they were subjective choices. Are there data in the literature to support it? Please add it.
Answer: These definitions were adapted from from Prostate Cancer Working Group 3 recommendations, reference [9] was added
-L100:pts (patients). "At time of this analysis" is not very clear. Please define a date.
Answer: Dataset was analyzed on September 25, 2022. This date was added
-In the table 1: There were mainly ISUP <or= 3, globally good pronostic. Please discuss it in the discussion.
Answer: majority of patients (74%) were affected by disease with ISUP score < 3, and likelihood of subclinical microscopic disease is lower in a similar population if compared to Gleason 8 or higher prostate cancer. For this reason, escalation in terms of local treatment might have been effective even in the absence of concomitant ADT administration. We added a comment in the discussion section.
-It would be good the have more figures, especially one showing the lesion of PET and MRI.
Answer: we added figure 1 with PSMA and MRI images together with treatment plan volumes and isodose lines.
- there were some typing mistakes such as L156 (NCT NCT), L163 (4,17-21,5), L166 (G2 adverse). It is definitively not exhaustive. Please correct them. Otherwise, it give a neglected impression and that's a shame as it is an interesting manuscript...
Answer: We performed a proofreading of the manuscript to correct typing errors.
Reviewer 3 Report
The authors present a paper about "Early results from a prospective trial investigating STereotactic sAlvage Radiotherapy for macroscopic prostate bed Recurrence after prostatectomy".
The topic is absolutely interesting and therefore it deserves to be adequately considered.
The introduction is well written and provides the ratioanle for the study.
Regarding the inofmation about the patients included it would be interesting to have further data:
a) first of all which is the % of patients staged with Choline and the % of patients staged with PSMA CT-PET?
b) how many patients did not perform MRI due to contraindications?
c) which is the median time from surgery to recurrence?
d) which is the definition of Nx used by the authors? (no lymph nodes removed or fewer that a predefined cut-off?)
e) please provide the complete ISUP classes and not only < and >3
f) was the SBRT treatment delivered avery other day?
g) did the patients receive any imaging after surgery (and before recurrence) to exclude the presence of residual prostate gland tissue?
h) which was the median PSA baseline after surgery (before recurrence)?
i) did the patients have a bladder catheter during the treatment?
Author Response
The authors present a paper about "Early results from a prospective trial investigating STereotactic sAlvage Radiotherapy for macroscopic prostate bed Recurrence after prostatectomy".
The topic is absolutely interesting and therefore it deserves to be adequately considered.
The introduction is well written and provides the ratioanle for the study.
Regarding the inofmation about the patients included it would be interesting to have further data:
- a) first of all which is the % of patients staged with Choline and the % of patients staged with PSMA CT-PET?
- b) how many patients did not perform MRI due to contraindications?
Answer: Macroscopic recurrence was detected by PSMA PET/CT, Choline PET/CT 15 (79%) and 4 (21%) patients, respectively. Five patients did not perform MRI confirmation. We added a comment in section 3
- c) which is the median time from surgery to recurrence?
Answer:Median time from surgery to recurrence was 37 months (IQR 21.7-124.5), we added a comment in the results section.
- d) which is the definition of Nx used by the authors? (no lymph nodes removed or fewer that a predefined cut-off?)
Answer:Nx was defined as no lymph nodes removed. We added a note to table 2.
- e) please provide the complete ISUP classes and not only < and >3
Answer: In table 2, We added Gleason Scores (3+3-3+4-4+3 etc) for each ISUP pattern, and added number of patients for each subclass
- f) was the SBRT treatment delivered avery other day?
Answer: Yes, SBRT was delivered every other day. We added a comment in section 2.2
- g) did the patients receive any imaging after surgery (and before recurrence) to exclude the presence of residual prostate gland tissue?
Answer: No, imaging was routinely performed after surgery and before recurrence. We added a comment in section 2.1
- h) which was the median PSA baseline after surgery (before recurrence)?
Answer: Median PSA at recurrence was 1.13 ng/ml (IQR 0.43-2.3 ng/ml). We added a comment in the results section.
- i) did the patients have a bladder catheter during the treatment?
Answer: no. We added a comment in section 2.2
Reviewer 4 Report
This is an early report of a prospective study aiming to enroll 90 patients with 2-year follow-up for their primary endpoint. At present, the authors have 19 patients with 3-month follow-up. Overall, I find it very difficult to make any inference or conclusion at this very early stage of study: 3-month PSA response after sRT provides very limited information in the long run.
There are some points for better clarity:
Methods:
The indication for CT/PET was not stated clearly. Was it done routinely at a certain point after prostatectomy? Or is it done only after BCR is confirmed?
Because the indication of CT/PET was not clear, the definition and indication of salvage RT here were vague. If CT/PET was positive in the absence of BCR (although the possibility should be low), was RT given anyway? And was it still defined as salvage in this setting?
Please give a reference for the criteria of BCR after salvage RT
At the time of enrollment (Mar, 2021), 6 months of concurrent ADT (usually done with LHRH agonist) should be the standard while delivering salvage RT. The authors might want to justify their study design of not using ADT, especially when the majority of patients are high risk.
Discussion:
The manuscript has one super long paragraph in Discussion. I would suggest a more structured arrangement. Some points of interest would be: the current status of PET-directed sRT in either local recurrence or oligometastasis, and comparison with other series not using ADT.
Author Response
This is an early report of a prospective study aiming to enroll 90 patients with 2-year follow-up for their primary endpoint. At present, the authors have 19 patients with 3-month follow-up. Overall, I find it very difficult to make any inference or conclusion at this very early stage of study: 3-month PSA response after sRT provides very limited information in the long run.
There are some points for better clarity:
Methods:
The indication for CT/PET was not stated clearly. Was it done routinely at a certain point after prostatectomy? Or is it done only after BCR is confirmed?
Answer: CT/PET was performed only after BCR was confirmed.We added a comment in section 2.1
Because the indication of CT/PET was not clear, the definition and indication of salvage RT here were vague. If CT/PET was positive in the absence of BCR (although the possibility should be low), was RT given anyway? And was it still defined as salvage in this setting?
Answer: As stated above, all patients were treated after biochemical recurrence, in a salvage setting, provided that a PSA >0.2 ng/ml was detected after surgery. Thus, no patient with positive imaging and absence of BCR could be enrolled.
Please give a reference for the criteria of BCR after salvage RT
Answer: definitions for biochemical relapse after SSRT were adapted from Prostate Cancer Working Group 3 recommendations, we added reference 9
At the time of enrollment (Mar, 2021), 6 months of concurrent ADT (usually done with LHRH agonist) should be the standard while delivering salvage RT. The authors might want to justify their study design of not using ADT, especially when the majority of patients are high risk.
Answer: Of course, some clinician may feel that short course concomitant ADT should be performed in a similar scenario. However, trials suggesting benefit for concomitant administration of ADT in this setting (e.g GETUG AFU 16 or RTOG 9601 [12,13]) were conducted in patients in whom methabolic imaging was not performed before salvage treatment, suggesting that part of the benefit could be related to spatial cooperation between local and systemic treatment. In our opinion, use of local treatment alone would be justified in a prospective trial conducted on patients with a more precise staging aimed to exclude subclinical disease outside the prostate bed. We added a comment in the discussion section
Discussion:
The manuscript has one super long paragraph in Discussion. I would suggest a more structured arrangement. Some points of interest would be: the current status of PET-directed sRT in either local recurrence or oligometastasis, and comparison with other series not using ADT.
Answer: We divided the discussion section in 5 subsection to be more easy to be read (Clinical Outcomes; ADT administration; Current status of PET-directed sRT; Dose escalated radiotherapy in salvage setting and comparison with other SSRT series and Potential advantages of SSRT ), according to the suggestion)
Round 2
Reviewer 4 Report
The manuscript is well written. However, I still think the conclusion is overreached based on the super early results.
I would suggest the title be modified to "Three-month PSA response and toxicity..." or "Three-month results..."
Author Response
The manuscript is well written. However, I still think the conclusion is overreached based on the super early results.
I would suggest the title be modified to "Three-month PSA response and toxicity..." or "Three-month results..."
Answer: we changed the title according to the suggestion